# Analysis of the coupling degree between regional logistics efficiency and economic development coordination

Na Li[1], Tianxing Ma[1], Xiaochun Deng[2]*

1 Chengdu Neusoft University, Chengdu, Sichuan, China, 2 China West Normal University, Nanchong, Sichuan, China

* 19115950185@163.com

**Data Availability Statement:** All relevant data are within the paper and its Supporting Information files.

**Funding:** The authors received no specific funding for this work.

## Abstract

This paper aims to study the relationship between regional logistics efficiency and economic development in 31 provinces of China and analyze their coupling coordination. To comprehensively evaluate the coordination between logistics and the economy, we introduced external indicators, such as carbon emissions, based on traditional evaluation indicators. We constructed an evaluation index system to coordinate regional logistics efficiency and economic development. The research approach used in this paper is the cross-DEA method, and data from 2010 to 2019 were selected for empirical calculation. The research findings indicate that Eastern and Northern regions of China show higher logistics efficiency, while Northwestern and Southwestern regions exhibit lower logistics efficiency. Coastal areas have relatively higher economic development levels compared to inland areas. Regarding the coupling coordination between logistics efficiency and economic development, different regions show temporal fluctuations and spatial disparities. Some regions demonstrate higher coordination between logistics efficiency and economic development, while others show lower coordination. Additionally, as the economy experiences rapid growth, logistics efficiency also improves, but the level of coordination varies among different provinces.

## I. Introduction

In today's context of globalization and regional economic integration, logistics plays a crucial role as a vital link that connects production and consumption, supporting economic development. An efficient logistics system has the potential to accelerate the flow of goods, lower logistics costs, and enhance production efficiency. Consequently, it contributes to stimulating economic growth and strengthening regional competitiveness.

However, in practical situations, some regions may need help with issues such as incomplete logistics networks, outdated transportation methods, and inadequate logistics information flow, leading to low logistics efficiency and impacting the speed and quality of economic development. On the other hand, the varying levels of economic development also impose higher demands on the logistics system, requiring it to adapt and support faster and more flexible economic growth. As a result, there are significant differences in logistics and economic

**Competing interests:** The authors have declared that no competing interests exist.

development among different regions. These disparities may influence the coordination and coupling degree between logistics efficiency and economic development, necessitating a regional-specific analysis to develop more targeted optimization measures.

Logistics efficiency refers to achieving high efficiency and high quality in logistics activities based on resource utilization and cost control. Common methods for measuring logistics efficiency include Data Envelopment Analysis (DEA), Stochastic Frontier Analysis (SFA), and Economic Efficiency Analysis. These methods can assess the efficiency level of the logistics system from different perspectives and help identify potential areas for improvement. Regional economic development involves various economic indicators such as GDP, employment rate, and industrial structure. Typically, statistical methods and economic models are used to evaluate the level and trends of regional economic development. Relevant studies have shown that improving logistics efficiency can promote economic growth, reduce transaction costs, and enhance production efficiency. Additionally, they emphasize the requirements of economic development levels on the logistics system, indicating that different levels of economic development may have an impact on logistics efficiency.

Currently, research mainly focuses on the relationship between the logistics efficiency of the logistics industry or companies and economic development. Still, there needs to be more exploration from a regional perspective regarding the connection between logistics development and economic growth. Additionally, when selecting evaluation indicators, researchers consider only internal influencing factors such as freight volume and goods turnover. However, the coordination between regional logistics efficiency and economic development is influenced by internal factors and external and regulatory factors. The existing research needs to consider the impact of external indicators such as energy consumption and carbon emissions on logistics development and economic growth.

Specifically, this study explores the relationship between regional logistics efficiency and economic development, particularly the coordination coupling degree. It considers the impact of external indicators, such as carbon emissions, on logistics development and economic growth. Based on this, a comprehensive evaluation index system for logistics efficiency and economic development is established, and the cross-DEA method is used to analyze data from 31 provinces in China empirically. The study aims to quantify the coordination degree between regional logistics efficiency and economic development and provide decision-makers with guidance on the impact of logistics on economic growth and development strategies. Additionally, this research will provide a scientific basis for comparing and evaluating different regions, promoting the synergistic development of logistics and the economy.

The paper is structured as follows: Section II will conduct a literature review. Section III will provide a detailed introduction to the research methods employed. Section IV will present the evaluation indicators and data sources. Section V will conduct an empirical analysis. Finally, Section VI will summarize the research findings.

## II. Literature reviews

### A. Definition and measurement methods of logistics efficiency

Logistics efficiency refers to achieving high efficiency and quality of logistics activities based on resource utilization and cost control. The logistics efficiency measurement methods commonly used in the literature include Data Envelopment Analysis (DEA), Stochastic Frontier Analysis (SFA) and Economic efficiency analysis. These methods can evaluate the efficiency level of logistics systems from different perspectives and help identify potential areas for improvement. Neto et al. (2007) developed a way to explore Pareto optimal solutions for business and the environment, which allows decision-makers to evaluate their preferred solution

through one of their most effective cognitive abilities—visual inspection [1]. Eleonora et al. (2015) proposed a detailed framework for designing sustainable and efficient logistics solutions for food industry zones and evaluating the resulting economic and environmental performance [2]. Zheng et al. (2020) used a combination of SBM method (one of the DEA methods) and hierarchical regression to study the efficiency and performance of regional logistics in China under carbon constraints, and explored the effects of external and internal factors on logistics efficiency, economic performance, and environmental performance in different regions and periods [3]. Luis et al. (2022) estimated the efficiency of 21 major airports in Europe during 2009–2014 using Data envelopment analysis [4]. Ramanathan (2007) used online customer rating data to explore how the relationship between logistics performance and customer loyalty is influenced by product risk characteristics and website efficiency [5]. Cavaignac et al. (2021) analyzed the production performance and efficiency determinants of 132 companies (Third-party logistics) in 2016 [6].

## B. Indicators and evaluation methods for regional economic development

Regional economic development involves a series of economic indicators, such as GDP, employment rate, industrial structure, etc. Researchers usually use statistical methods and financial models to evaluate the level and trend of regional economic development. For example, commonly used evaluation methods include econometric models, spatial econometric techniques, and input-output models. Wu et al. (2020) introduced a super PEBM model and window analysis technique based on environmental DEA technology to systematically analyze the sequence evolution, spatial differentiation, and dynamic evolution of GEE in various provinces of China from 2008 to 2017 [7]. Deng et al. (2013) used structural equation modelling to establish a model of the impact of port supply, port demand, and port factor value-added activities on the regional economy [8]. Based on the Gini coefficient construction method with continuous distribution function, Shu and Xiong (2018) expanded the traditional Gini coefficient of income distribution to the environmental Gini coefficient, which is used to measure the economic and ecological balance of regional industrial economic development when the distribution sample size is limited [9]. Zhong et al. (2010) used the DEA input-oriented CRS and VRS models to evaluate the performance of industrial enterprises' R&D investment in 30 provinces of China based on the first China Economic Census [10]. Khan, S.A.R., Yu, Z. & Umar, M. (2022) The results of FMOLS and DOLS tests on data from 12 cities show that the above factors are crucial in the transition and have a positive significance in promoting green total factor productivity and reducing non-renewable energy consumption and environmental pollution [11].

## C. Relationship between logistics efficiency and economic development

Many studies have explored the relationship between logistics efficiency and economic development. Some studies have shown that improving logistics efficiency can promote economic growth, reduce transaction costs, and improve production efficiency. Other studies emphasize the requirements of economic development levels for logistics systems, pointing out that differences in economic development levels may impact logistics efficiency. These studies provide essential theoretical and empirical foundations for understanding the interrelationship between logistics and the economy. Chao et al. (2021) examined the causal relationship between logistics infrastructure and China's economic development. Therefore, the Granger causality between logistics infrastructure and economic development indicators is tested under the VAR and VECM frameworks [12]. Ruhe et al. (2022) constructed an evaluation index system, determined weights using the entropy weight method, and systematically

studied the coupling relationship between China's cold chain logistics and the Chinese economy from 2010 to 2019 based on the coupled coordination degree model (CCDM) [13]. Based on relevant data from 11 cities in Jiangxi Province, Weihua et al. (2022) selected indicators related to high-quality development and green logistics efficiency in the context of a low-carbon environment. Through correlation analysis, factor analysis, and high selection coefficient, green logistics efficiency evaluation indicators were established, and comprehensive technical efficiency was measured. The pure technical efficiency and scale efficiency of 11 cities in Jiangxi Province were analyzed using a three-stage DEA model [14]. Zhang et al. (2020) used coupling theory to construct a coupling coordination model for economic development, logistics development, and ecological environment among China's four major economic regions. They calculated the financial development index, logistics development index, and environmental background comprehensive index of 30 provinces and cities in China [15]. Lean et al. (2014) used the latest available data and advanced econometrics to examine the short-term and long-term causal relationship between logistics development and economic growth [16]. Zhou et al. (2020) proposed a system of urban logistics and economic evaluation indicators obtained through literature review and field investigations and served as the basis for quantitative evaluation [17]. Khan S A R, Yu Z, Umar M, et al (2022) used fixed and random effects models to test the impact of advanced logistics and renewable energy on carbon-free economic growth and international agricultural products. The results show they are crucial in boosting the economy and global agricultural products [18].

## D. Contributions

In summary, existing research predominantly concentrates on the correlation between logistics efficiency and economic development within the logistics industry or logistics enterprises, with limited exploration of the relationship between logistics development and regional economic growth. Furthermore, when selecting evaluation indicators, the focus is typically on internal factors such as freight volume and turnover. However, the efficiency of regional logistics and the coordination with economic development is influenced not only by internal factors but also by external and regulatory factors. The current research must consider the impact of external indicators, such as energy consumption and carbon emissions, on logistics development and economic growth.

## III. Evaluation method

This paper adopts the DEA cross-efficiency model and the coupling coordination degree calculation formula to comprehensively evaluate the efficiency and coordination degree of regional logistics and economic development. The DEA cross-efficiency model quantitatively assesses the efficiency of regional logistics and economic development using self-assessment values and cross-efficiency as comprehensive indicators. The coupling coordination degree calculation formula considers the cross-efficiency values of logistics and economic development to measure the coordination degree between logistics and economic development. The specific steps of the evaluation method are as follows: first, applying the DEA cross-efficiency model to calculate the efficiency level of regional logistics and economic development, and then using the coupling coordination degree calculation formula to calculate the coordination degree between logistics and economic development.

### A. DEA cross efficiency

Doyle and Green (1994) proposed a DEA cross-efficiency model to solve DEA decision units' sorting and comparison problems effectively [19]. This model goes beyond the scope of only

considering individual selection preferences of DMUs but comprehensively incorporates factors that affect the weight changes of other DMUs. By using self-evaluation values and cross-efficiency as comprehensive indicators to measure the efficiency of DMUs, the cross-efficiency model solves the problem of incompatibility caused by inconsistent input and output weights in the CCR model. Based on that, this paper uses the DEA cross-efficiency to take into account the interactions between economic development and logistics efficiency among these regional areas.

The principle of the DEA cross efficiency model is that there are $n$ self-evaluation decision units $DMU_j$. Where $j = 1,2,...,n$. The input and output vectors of each decision-making unit are: $x_j = (x_{1j}, x_{2j},...,x_{mj})$, $y_j = (y_{1j}, y_{2j},...,y_{mj})$, solving constrained objective programming problems:

$$maximum \frac{\sum_{r=1}^{s} u_r y_{rj}}{\sum_{i=1}^{m} v_r x_{rj}} \tag{1}$$

s.t.

$$\frac{\sum_{r=1}^{s} u_r y_{rj}}{\sum_{i=1}^{m} v_r x_{rj}} \leq 1 \; j = 1, 2, 3, \ldots, n \tag{2}$$

$$u_r \geq 0 \; r = 1, 2, 3, \ldots, s \tag{3}$$

$$v_i \geq 0 \; i = 1, 2, 3, \ldots, m \tag{4}$$

where, $x_j = (x_{1j}, x_{2j},...,x_{mj})$ and $y_j = (y_{1j}, y_{2j},...,y_{mj})$ represents the $m$ input and $s$ output vectors of each $j$th decision unit. $v = (v_1, v_2,...,v_m)^T$ and $u = (u_1, u_2,...,u_s)^T$ represent the weight vectors of $m$ inputs and $s$ outputs, respectively. $DMU_j = (j = 1,2,...,n)$ represents the $j$th self-evaluation decision unit.

Solve the constrained goal programming problem (1)-(4) to obtain the optimal weights of the $j$th decision-making unit as $v^* = (v_{1j}^*, v_{2j}^*, \ldots, v_{mj}^*)^T$ and $u^* = (u_{1j}^*, u_{2j}^*, \ldots, u_{mj}^*)^T$. Substitute the optimal weights into the input and output indicators of the $j$th decision-making unit to obtain the corresponding scale efficiency $G_j$ of the $j$th decision-making unit as a self-evaluation value:

$$G_j = \frac{\sum_{r=1}^{s} u_{rj}^* y_{rj}}{\sum_{i=1}^{m} v_{ij}^* x_{ij}} j = 1, 2, 3, \ldots, n \tag{5}$$

The optimal solution $u_i^*, v_i^*$ of the model are not unique. It is important to ensure that every $DMU_i$ is under the condition of obtaining the maximum efficiency value, and the other $DMU_i$ obtain the smallest possible cross efficiency value. The cross DEA model is established as follows:

$$\begin{cases} max y_i^T u = E_i \\ s.t. y_i^T u \leq x_j^T v (1 \leq j \leq n) \\ \quad x_j^T v = 1 \\ \quad u \geq 0, v > 0 \end{cases} \tag{6}$$

$$\begin{cases} min y_k^T u_i \\ s.t. y_i^T u \leq x_j^T v (1 \leq j \leq n) \\ \quad x_k^T v = 1 \\ \quad y_i^T u = E_i x_i^T v \end{cases} \tag{7}$$

Obtain the cross-efficiency value $E_{jd}$ of the $j$th and $d$th decision units from the model (8):

$$E_{jd} = \frac{\sum_{r=1}^{s} u_{rd}^* y_{rj}}{\sum_{i=1}^{m} v_{ij}^* x_{ij}} j = 1, 2, \ldots, n; d = 1, 2, \ldots, n \tag{8}$$

In this context, $E_{jd}$ d represents the most effective value of the decision unit when $j = d$. The other elements are cross-evaluation values, which are obtained by taking the average of $E_{jd}$. This process yields the cross-efficiency of the $j$th decision unit.

$$\bar{E}_j = \frac{1}{n} \sum_{d=1}^{n} E_{jd} \tag{9}$$

## B. Evaluation of coupling coordination degree

The calculation formula for the coupling degree C is as follows:

$$C = 2 \left[ \frac{u_1 u_2}{(u_1 + u_2)^2} \right]^{1/2} \tag{10}$$

Where $u_1$ represents the cross-efficiency of logistics and $u_2$ represents the cross-efficiency of economic development. Since cross-efficiency values range between 0 and 1, excluding values of 0 or 1, using DEA cross-efficiency values of logistics and economic development levels ensures that the calculated coupling degree C remains between 0 and 1, and each value is distinct from one another.

To analyze the impact of cross-efficiency between logistics and economic development on coupling coordination, we calculate the comprehensive coordination index $T$ and the static coupling coordination development degree $D$.

$$T = \alpha u_1 + \beta u_2 \tag{11}$$

$$D = \sqrt{C * T} \tag{12}$$

Where $\alpha$ and $\beta$ are undetermined parameters, typically set to $\alpha = \beta = 0.5$. The value of $D$ lies between 0 and 1, where a higher $D$ value indicates a higher level of coordinated development in the system. The coordination of the logistics and economic development system is classified into five levels based on $D$. The specific criteria can be found in **Table 1**.

## IV. Evaluation indicators and data sources

### A. Establishment of indicator system

Based on comprehensive and scientifically reasonable indicator principles, this paper focuses on promoting low-carbon and green development of regional logistics and economic systems. It comprehensively refers to the indicator systems of multiple scholars. It simplifies them as much as possible, constructing an evaluation indicator system for the coordination coupling degree between regional logistics efficiency and economic development, as shown in **Table 2**.

**Table 1. Coordination level classification standards.**

| Range of D values | (0.0,0.2] | (0.2,0.4] | (0.4,0.6] | (0.6,0.8] | (0.8,1.0] |
|---|---|---|---|---|---|
| Coordination Level | Severely Inconsistent | Relatively Inconsistent | Basically Coordinated | Relatively Coordinated | Highly Coordinated |

**Table 2. Regional logistics and economic development indicator system.**

| Regional Logistics and Economic Development | Indicator System | Basic Indicators |
|---|---|---|
| Logistics System | Logistics System | Fixed assets investment in transportation ($x_{11}$) |
| | | Number of logistics practitioners ($x_{12}$) |
| | | Logistics energy consumption ($x_{13}$) |
| | Beneficial Output Indicators | Cargo transportation volume ($y_{11}$) |
| | | Turnover volume of goods ($y_{12}$) |
| | | Logistics carbon emissions ($y_{13}$) |
| Economic System | Resource Input Indicators | Resource Input Indicators ($x_{21}$) |
| | | Total retail sales of social consumer goods ($x_{22}$) |
| | | Proportion of logistics personnel ($x_{23}$) |
| | | Energy consumption per unit of GDP ($x_{24}$) |
| | Beneficial Output Indicators | Gross domestic product ($y_{21}$) |
| | | Fiscal revenue ($y_{22}$) |
| | | Output value of the logistics industry ($y_{23}$) |
| | | Clean energy output ($y_{24}$) |

For the logistics system, input indicators include fixed assets investment in transportation, the number of logistics practitioners, and logistics energy consumption. Output indicators include cargo transportation volume, turnover volume of goods, and logistics carbon emissions. For the economic system, input indicators include per capita disposable income of residents, total retail sales of social consumer goods, the proportion of logistics personnel, and energy consumption per unit of GDP. Output indicators include gross domestic product, fiscal revenue, the output value of the logistics industry, and clean energy output. Furthermore, the logistics and economic system indicator systems include indicators of different dimensions and magnitudes. To eliminate their impact on the analysis, a dimensionless method is used to standardize the data, ensuring that all data fall within the range of [0, 1].

### 1) Logistics system input-output indicators

Environmental protection has become a global concern with the increasing prominence of environmental issues. As an integral part of economic development and goods circulation, the logistics system, directly and indirectly, impacts the environment. By considering environmental factors, it is possible to assess the logistics system's influence on greenhouse gas emissions, energy consumption, and air and water pollution and propose improvements and optimization measures to reduce the environmental burden. Additionally, ecological investments can promote the sustainable development of the logistics system, enhance resource utilization efficiency and energy efficiency, and drive the construction of green supply chains to meet consumers' demand for environmentally-friendly products and services. The specific content of the relevant input-output indicators is as follows [20].

Fixed assets investment in transportation ($x_{11}$): Fixed assets investment in transportation refers to the capital input used for the acquisition, construction, and maintenance of transportation infrastructure and vehicles in the field of transportation. These fixed assets include transportation infrastructure such as roads, railways, aviation, waterways, as well as transportation vehicles such as trucks and trains. The scale and quality of fixed assets investment in

transportation directly impact the level of transportation network completeness, transportation efficiency, and safety.

Number of logistics practitioners ($x_{12}$): The number of logistics practitioners refers to the statistical count of individuals engaged in various professions within the logistics industry. This includes individuals involved in logistics transportation, warehousing, distribution, supply chain management, logistics information systems, and related occupations. The number of logistics practitioners is an important indicator for measuring the scale and development of the logistics industry. It plays a significant role in assessing employment conditions, labor demands, and industry development within the logistics sector.

Logistics energy consumption ($x_{13}$): Logistics energy consumption refers to the amount of energy consumed in logistics activities, including fuel consumption during transportation, energy consumption in warehouse facilities, and energy consumption in logistics information systems, among others. Considering environmental factors, reducing energy consumption in the logistics industry is a key factor in promoting sustainable development.

Cargo transportation volume ($y_{11}$): Cargo transportation volume refers to the total quantity of goods transported through the logistics system within a certain period of time. It is an important indicator for measuring the scale of logistics activities and the level of development in the logistics industry. Considering environmental factors, reducing the growth rate of cargo transportation volume and optimizing the structure of cargo transportation are crucial for reducing energy consumption and environmental impacts.

Turnover volume of goods ($y_{12}$): Turnover volume of goods refers to the quantity of goods that circulate and exchange through the logistics system within a specific period of time. It is an important indicator for measuring the efficiency of logistics transportation and storage activities. The turnover volume of goods can be used to assess the operational efficiency of the logistics system, the utilization rate of logistics facilities, and the speed of goods circulation.

Logistics carbon emissions ($y_{13}$): Logistics carbon emissions refer to the quantity of greenhouse gases, such as carbon dioxide ($CO2$), emitted as a result of energy consumption and emissions during transportation in logistics activities. Considering environmental factors, reducing logistics carbon emissions is one of the important goals in promoting low-carbon economic development and addressing climate change.

## 2) Economic system input-output indicators

Traditional economic indicators such as Gross Domestic Product (GDP) mainly focus on economic growth while overlooking environmental quality and resource consumption issues. However, environmental degradation and resource scarcity pose substantial threats to economic output and the smooth functioning of logistics. By considering environmental factors, input-output indicators can better measure the impact of economic activities on the environment and provide a more accurate assessment of financial sustainability. The specific content of the relevant input-output indicators is as follows [21].

Per capita disposable income of residents ($x_{21}$): Per capita disposable income of residents refers to the total income available to an average resident for discretionary spending within a certain period of time. It is an important indicator for measuring people's economic living standards and purchasing power. Per capita disposable income of residents has significant reference value for the government in formulating social welfare policies, adjusting taxation, and social security systems. It reflects the economic strength and quality of life of residents and to some extent, indicates the level of economic fairness and sustainable development within society.

Total retail sales of consumer goods ($x_{22}$): Total retail sales of consumer goods refer to the total value of all goods and services sold in a country or region within a certain period of time.

The growth of total retail sales of consumer goods is often considered as an indicator of economic vitality and market demand. It is of great significance for evaluating economic growth, consumption trends, and the level of business prosperity in a region. The data on total retail sales of consumer goods provide important references for governments, businesses, and investors in formulating strategic decisions, market forecasts, and sales strategies.

Proportion of logistics personnel ($x_{23}$): The proportion of logistics personnel refers to the ratio between the number of people engaged in the logistics industry and the total labor force within a specific economic system. This indicator can measure the importance and scale of the logistics industry within the economy. The proportion of logistics personnel can reflect the contribution of the logistics industry to economic development and the movement of goods. A higher proportion of logistics personnel usually indicates greater employment opportunities and economic value of the logistics industry within the economic system, while a lower proportion may suggest a relatively smaller logistics industry or the adoption of more efficient logistics management practices within the economic system.

Unit GDP energy consumption ($x_{24}$): Unit GDP energy consumption refers to the total amount of energy consumed in producing one unit of Gross Domestic Product (GDP) within a specific economic system. This indicator is used to measure the energy efficiency and resource utilization of economic activities. A low value of unit GDP energy consumption typically indicates that the economy consumes less energy to achieve a certain level of output, reflecting energy-saving and environmental-friendly practices. Conversely, a higher unit GDP energy consumption implies that the economy consumes more energy for the same level of output, indicating potential resource waste and environmental pressure.

Gross Domestic Product (GDP) ($y_{21}$): Gross Domestic Product (GDP) refers to the total market value of all final goods and services produced within a country or region during a specific period of time. It is an important indicator for measuring the economic size and output level of a country. GDP encompasses various economic activities, including private consumption, government spending, investment, and net exports. By calculating and comparing GDP, one can assess the economic growth rate, industrial structure, distribution of economic activities, and the level of economic well-being [22].

Fiscal revenue ($y_{22}$): Fiscal revenue refers to the total amount of funds that a government obtains from various sources during a specific period of time. These sources include taxes, non-tax revenues, government debt, sponsorships, and donations, among others. Fiscal revenue is one of the necessary sources of funds for government operations and the provision of public services. The level and composition of fiscal revenue can reflect the economic situation of a country, the effectiveness of tax policies, the scale and distribution of economic activities, as well as the financial health of the government.

Logistics industry output value ($y_{23}$): Logistics industry output value refers to the total value created by the logistics industry in a country or region during a specific period of time. It represents the contribution and scale of the logistics industry to the economy. The calculation of logistics industry output value typically includes the total value created by logistics enterprises engaged in transportation, warehousing, packaging, distribution, and other logistics activities. This indicator can reflect the importance of the logistics industry in supply chain management and goods circulation, as well as its contribution to economic growth and market development.

Clean energy output ($y_{24}$): Clean energy output refers to the total amount of energy generated through renewable energy and other clean energy technologies in a country or region during a specific period of time. Clean energy includes renewable energy sources such as solar energy, wind energy, hydropower, biomass, as well as low-carbon and low-emission energy forms such as nuclear power. Increasing clean energy output is significant for reducing

reliance on traditional fossil fuels, lowering carbon emissions, addressing climate change, and protecting the environment.

## B. Data sources

A region usually refers to a certain geographical space. It has a certain area, shape, scope or boundary. Due to the vastness of China's territory, there are various types of geographic regions analyzed from multiple perspectives: topographical, climatic, human, economic and political. In general, China is usually divided into seven geographic regions. This paper focuses on the degree of coordinated coupling between regional logistics efficiency and economic development, with economic geography as the primary basis, so this paper follows this regional division into the following seven regions:

- Northeast Region (Liaoning, Heilongjiang, Jilin)

- North China Region (Beijing, Tianjin, Hebei, Shanxi, Inner Mongolia)

- East China Region (Shanghai, Jiangsu, Zhejiang, Anhui, Fujian, Jiangxi, Shandong)

- Central China Region (Henan, Hubei, Hunan)

- South China Region (Guangdong, Guangxi, Hainan)

- Southwest China Region (Guizhou, Sichuan, Yunnan, Chongqing, Tibet)

- Northwest China Region (Shaanxi, Gansu, Qinghai, Ningxia, Xinjiang)

This study is based on relevant data from 31 provinces in China (excluding Hong Kong, Macau, and Taiwan) from 2010 to 2019. The relevant data mainly comes from the "China Statistical Yearbook," "China Energy Statistical Yearbook," "China Environmental Statistical Yearbook," statistical yearbooks of 31 provinces, and statistical bulletins on the national economic and social development of each province. In cases where there are inconsistencies in the data from different yearbooks, the data published in the "China Statistical Yearbook" and on the website of the National Bureau of Statistics are used as the primary source.

The original data for carbon emissions primarily comes from the CEDAs database, which provides energy consumption and carbon emission inventories for various industries and provinces in China from 2001 to 2018. This database is currently the most authoritative statistical data for calculating carbon emissions in China. The missing original data for 2019 and carbon emissions were supplemented using interpolation methods.

## V. Empirical analysis

### A. Evaluation of regional logistics efficiency

The comprehensive logistics efficiency of different regions in China from 2010 to 2019 is shown in Table 3. It can be seen that the logistics efficiency of other areas of China showed fluctuating characteristics during the inspection period. From the perspective of time fluctuation, the logistics efficiency in East China and North China shows a stable trend of high efficiency, and the average logistics efficiency from 2010 to 2019 fluctuates at a high level above 0.9. The average logistics efficiency in South China fluctuated around 0.8 between 2010 and 2019. The average logistics efficiency in Central China has shown an increasing trend year by year from 2010 to 2019. The average logistics efficiency in the Northeast, Southwest, and Northwest regions fluctuated at low levels between 2010 and 2019, but there were differences. Among them, the average logistics efficiency in the Northeast region showed a significant increase after 2015, the average logistics efficiency in the Southwest region showed a significant

**Table 3. Comprehensive logistics efficiency in different regions of China from 2010 to 2019.**

| Region | 2010 | 2011 | 2012 | 2013 | 2014 | 2015 | 2016 | 2017 | 2018 | 2019 | Mean value |
|---|---|---|---|---|---|---|---|---|---|---|---|
| North China | 0.980 | 0.944 | 0.966 | 0.974 | 0.974 | 0.924 | 0.941 | 0.937 | 0.981 | 0.990 | 0.961 |
| Northeast region | 0.703 | 0.650 | 0.621 | 0.624 | 0.688 | 0.746 | 0.753 | 0.716 | 0.698 | 0.703 | 0.690 |
| East China | 0.937 | 0.980 | 0.973 | 0.955 | 0.937 | 0.964 | 0.985 | 0.967 | 0.972 | 0.980 | 0.965 |
| Central China | 0.714 | 0.761 | 0.784 | 0.759 | 0.745 | 0.781 | 0.795 | 0.852 | 0.810 | 0.824 | 0.783 |
| South China | 0.782 | 0.818 | 0.762 | 0.792 | 0.817 | 0.833 | 0.772 | 0.835 | 0.841 | 0.796 | 0.805 |
| Southwest region | 0.596 | 0.520 | 0.571 | 0.552 | 0.576 | 0.525 | 0.626 | 0.610 | 0.638 | 0.632 | 0.585 |
| Northwest region | 0.566 | 0.723 | 0.722 | 0.696 | 0.689 | 0.680 | 0.651 | 0.583 | 0.604 | 0.623 | 0.654 |
| National average | 0.754 | 0.771 | 0.771 | 0.765 | 0.775 | 0.779 | 0.789 | 0.786 | 0.792 | 0.793 | 0.777 |

improvement after 2016, and the average logistics efficiency in the Northwest region showed a small peak between 2011 and 2015. Overall, the logistics efficiency in the different areas of China offers a pattern of "East China＞North China＞South China＞Central China＞Northeast＞Northwest＞Southwest."

Further analyze the spatiotemporal evolution characteristics of logistics cross efficiency in seven major regions of China from 2010 to 2019, including North China, Northeast China, East China, Central China, South China, Southwest China, and Northwest China (see Fig 1). Significant differences exist in the comprehensive efficiency development level of logistics in different regions of China, and regional development is uneven. Among them, the average complete efficiency of North China and East China has consistently remained above 0.9, far higher than the national average. It is in a leading position in the region. This is mainly due to the rapid growth of the logistics economy, high technological innovation, and practical green development in these regions, with low carbon emissions. The average value of comprehensive efficiency in South China and Central China has always been within the range of 0.7–0.8 and is also at a high level. Mainly due to the level of economic development and regional logistics and transportation construction, the efficiency of logistics development in these two regions has improved rapidly. The average comprehensive efficiency of the Northeast region hovers between 0.6 and 0.7, and there is still room for further improvement, mainly due to the low efficiency of the regional logistics scale and the mismatch between the regional logistics scale and logistics input-output. The average comprehensive efficiency of the southwest and north-west regions ranges from 0.5 to 0.6, which is relatively backward nationwide. This indicates that these two regions may have excessive investment, high energy consumption, high carbon emissions, and low logistics output.

## B. Evaluation of regional economic development efficiency

The economic development efficiency of different regions in China from 2010 to 2019 is shown in **Table 4** and **Fig 2**. From 2010 to 2019, China's economic development level showed a trend of fluctuating development and stable rise. From a regional perspective, the average levels of high-quality economic growth in the seven major regions of North China, Northeast China, East China, Central China, South China, Southwest China, and Northwest China from 2010 to 2019 were 0.389, 0.329, 0.428, 0.377, 0.356, 0.301, and 0.191, respectively. According to the comparison results between the whole sample and seven regions, the high-quality economic development level of East China, South China, Central China, and North China is significantly higher than that of the other three regions and the national average. It has always been in a leading position. This is due to the favorable location and resource advantages of these regions, relatively optimized industrial structure, and the gathering of high-end factors

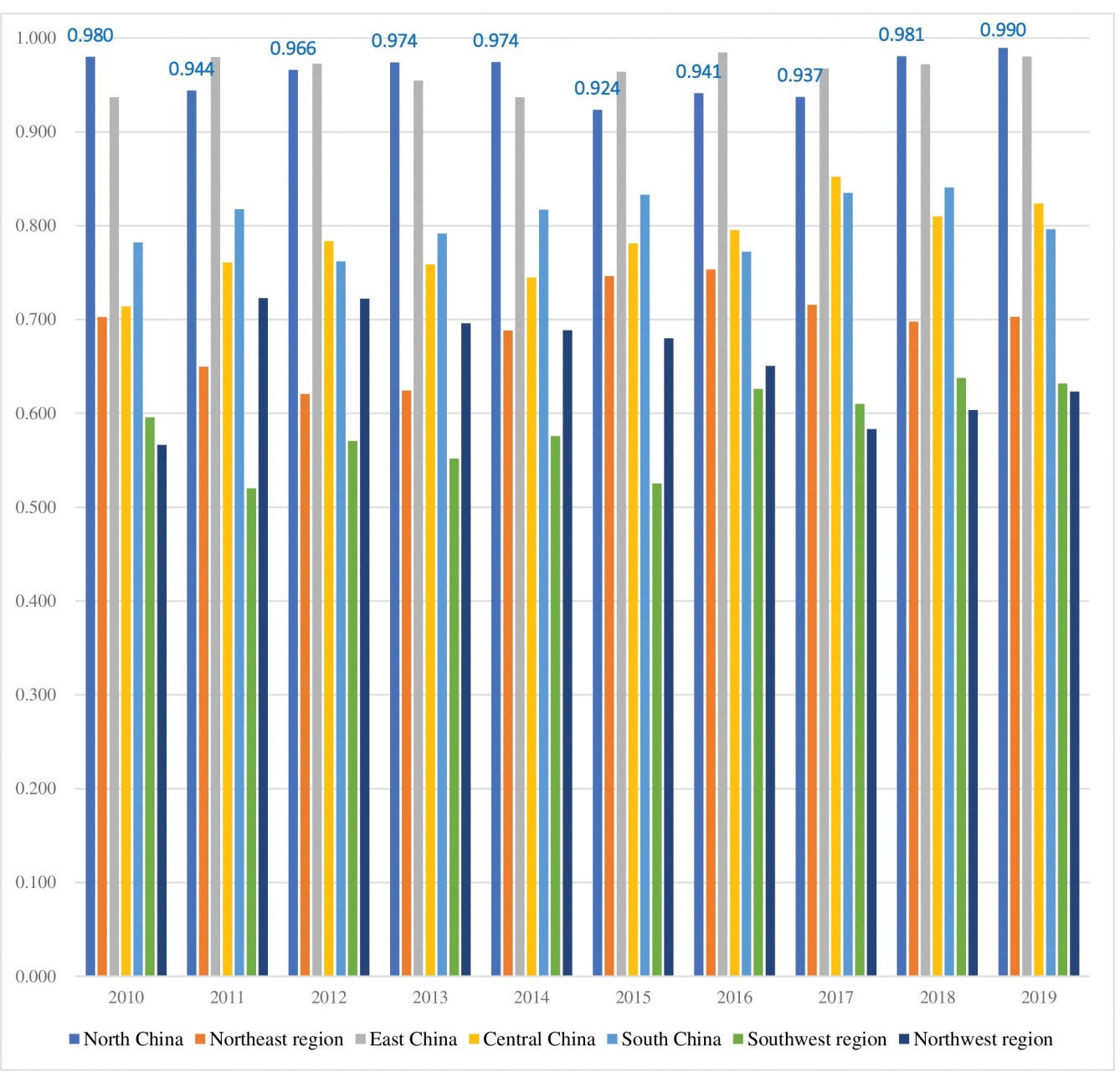

**Fig 1. Changes in average logistics comprehensive efficiency in different regions of China from 2010 to 2019.**

such as advanced technology, professional talents, and digital information, making these regions more vibrant and developed economically than others [23].

Further analysis of the spatiotemporal evolution characteristics of economic development efficiency in seven major regions of China from 2010 to 2019, including North China, Northeast China, East China, Central China, South China, Southwest China, and Northwest China, reveals that the overall level of regional economic development in China continued to grow during the inspection period, transitioning from rapid growth to uniform growth. This may be related to the adjustment of China's economic development strategy. Under the guidance of central policies, innovation-driven and green development are seen as essential indicators of

**Table 4. Economic development efficiency of different regions in China from 2010 to 2019.**

| Region | 2010 | 2011 | 2012 | 2013 | 2014 | 2015 | 2016 | 2017 | 2018 | 2019 | Mean value |
|---|---|---|---|---|---|---|---|---|---|---|---|
| North China | 0.397 | 0.398 | 0.400 | 0.396 | 0.394 | 0.388 | 0.388 | 0.385 | 0.374 | 0.378 | 0.390 |
| Northeast region | 0.340 | 0.342 | 0.341 | 0.342 | 0.339 | 0.339 | 0.325 | 0.315 | 0.306 | 0.310 | 0.330 |
| East China | 0.413 | 0.416 | 0.418 | 0.427 | 0.434 | 0.444 | 0.447 | 0.449 | 0.415 | 0.415 | 0.428 |
| Central China | 0.365 | 0.367 | 0.370 | 0.380 | 0.381 | 0.382 | 0.383 | 0.385 | 0.382 | 0.384 | 0.378 |
| South China | 0.339 | 0.343 | 0.346 | 0.356 | 0.357 | 0.358 | 0.363 | 0.363 | 0.368 | 0.372 | 0.356 |
| Southwest region | 0.256 | 0.266 | 0.277 | 0.297 | 0.309 | 0.312 | 0.320 | 0.322 | 0.321 | 0.326 | 0.301 |
| Northwest region | 0.177 | 0.181 | 0.188 | 0.190 | 0.192 | 0.193 | 0.195 | 0.195 | 0.196 | 0.205 | 0.191 |
| National average | 0.327 | 0.330 | 0.334 | 0.341 | 0.344 | 0.345 | 0.346 | 0.345 | 0.338 | 0.341 | 0.339 |

China's economic transition from high-speed growth to high-quality development. Therefore, the development focus of different regions in China has further shifted from economic operation to growth momentum and ecological environment, and the development benefits brought

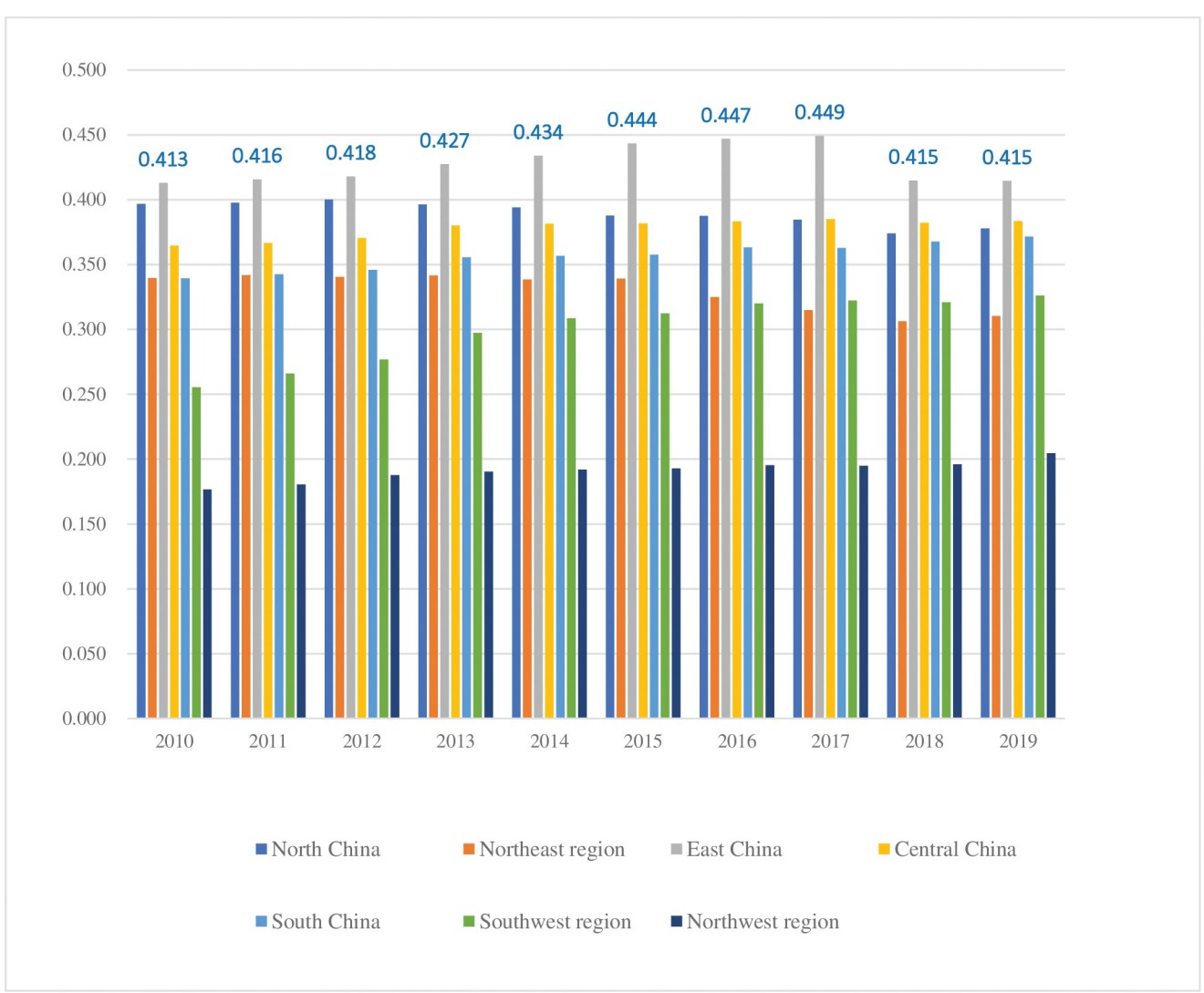

**Fig 2. Changes in average economic development efficiency in different regions of China from 2010 to 2019.**

by these two aspects often have a certain lag, leading to a slowdown in the economic development level of different regions in China from 2010 to 2019. With the continuous development of China's economy and society, the accumulation, scale, and radiation effects of cities and towns are increasingly strengthening. China's economic structure will continue to adjust, industries will be optimized and upgraded, and resources will be effectively utilized, thereby promoting the economic development efficiency of various regions to show a growth trend. However, there are still significant regional differences, and the development level will gradually decrease from coastal areas such as East China, North China, and South China to inland areas such as Southwest and Northwest.

## C. Analysis of the coupling degree between regional logistics efficiency and economic development coordination

Based on the evaluation of regional logistics efficiency and regional economic development efficiency, the coupling coordination degree model calculation method was used to calculate the coupling degree of logistics efficiency and economic development coordination in different regions of China from 2010 to 2019, as shown in **Table 5** and **Fig 3**.

It can be seen that the coordination and coupling between logistics efficiency and economic development in different regions of China exhibit temporal fluctuations and spatial differences. Specifically, the coordination degree of logistics efficiency and economic development coupling in North China showed a fluctuating downward trend from 2010 to 2019, from 0.618 in 2010 to 0.509 in 2019. The coordination coupling between logistics efficiency and economic development in the Northeast region also shows a fluctuating downward trend, from 0.457 in 2010 to 0.322 in 2019. The coordination coupling degree between logistics efficiency and economic development in East China fluctuates around 0.6. The coupling degree of logistics efficiency and economic development coordination in Central China is characterized by first rising and then falling from 0.445 in 2010 to 0.470 in 2013 and then falling to 0.439 in 2019. The coordination coupling degree between logistics efficiency and economic development in South China is relatively high all year round, with an average value of 0.558 from 2010 to 2019. The coordination coupling degree of logistics efficiency and economic development in southwest China is similar to that in Central China, which also shows the characteristics of first rising and then falling from 0.466 in 2010 to 0.491 in 2013 and then falling to 0.458 in 2019. The coordination coupling between logistics efficiency and economic development in the northwest region is consistently low, with an average value of 0.358 from 2010 to 2019.

**Table 6** shows the regional division based on different coordination levels from 2010 to 2019. It can be seen that the coordination coupling between logistics efficiency and economic development in Northeast China was at the basic coordination level from 2010 to 2014, and after 2015, it was at a lower level of coordination. This indicates that there has been a

**Table 5. Coordination between logistics efficiency and economic development efficiency in different regions of China from 2010 to 2019.**

| Region | 2010 | 2011 | 2012 | 2013 | 2014 | 2015 | 2016 | 2017 | 2018 | 2019 | Mean value |
|---|---|---|---|---|---|---|---|---|---|---|---|
| North China | 0.608 | 0.563 | 0.588 | 0.538 | 0.574 | 0.504 | 0.506 | 0.485 | 0.470 | 0.509 | 0.534 |
| Northeast region | 0.457 | 0.443 | 0.436 | 0.429 | 0.413 | 0.282 | 0.397 | 0.305 | 0.302 | 0.322 | 0.379 |
| East China | 0.610 | 0.590 | 0.587 | 0.590 | 0.541 | 0.560 | 0.513 | 0.519 | 0.534 | 0.541 | 0.559 |
| Central China | 0.446 | 0.465 | 0.471 | 0.470 | 0.445 | 0.343 | 0.347 | 0.364 | 0.438 | 0.439 | 0.423 |
| South China | 0.598 | 0.605 | 0.615 | 0.611 | 0.517 | 0.495 | 0.490 | 0.518 | 0.573 | 0.557 | 0.558 |
| Southwest region | 0.466 | 0.497 | 0.483 | 0.491 | 0.426 | 0.396 | 0.422 | 0.396 | 0.431 | 0.458 | 0.447 |
| Northwest region | 0.351 | 0.332 | 0.335 | 0.310 | 0.392 | 0.356 | 0.355 | 0.354 | 0.392 | 0.403 | 0.358 |
| National average | 0.505 | 0.499 | 0.502 | 0.491 | 0.472 | 0.420 | 0.433 | 0.420 | 0.449 | 0.461 | 0.465 |

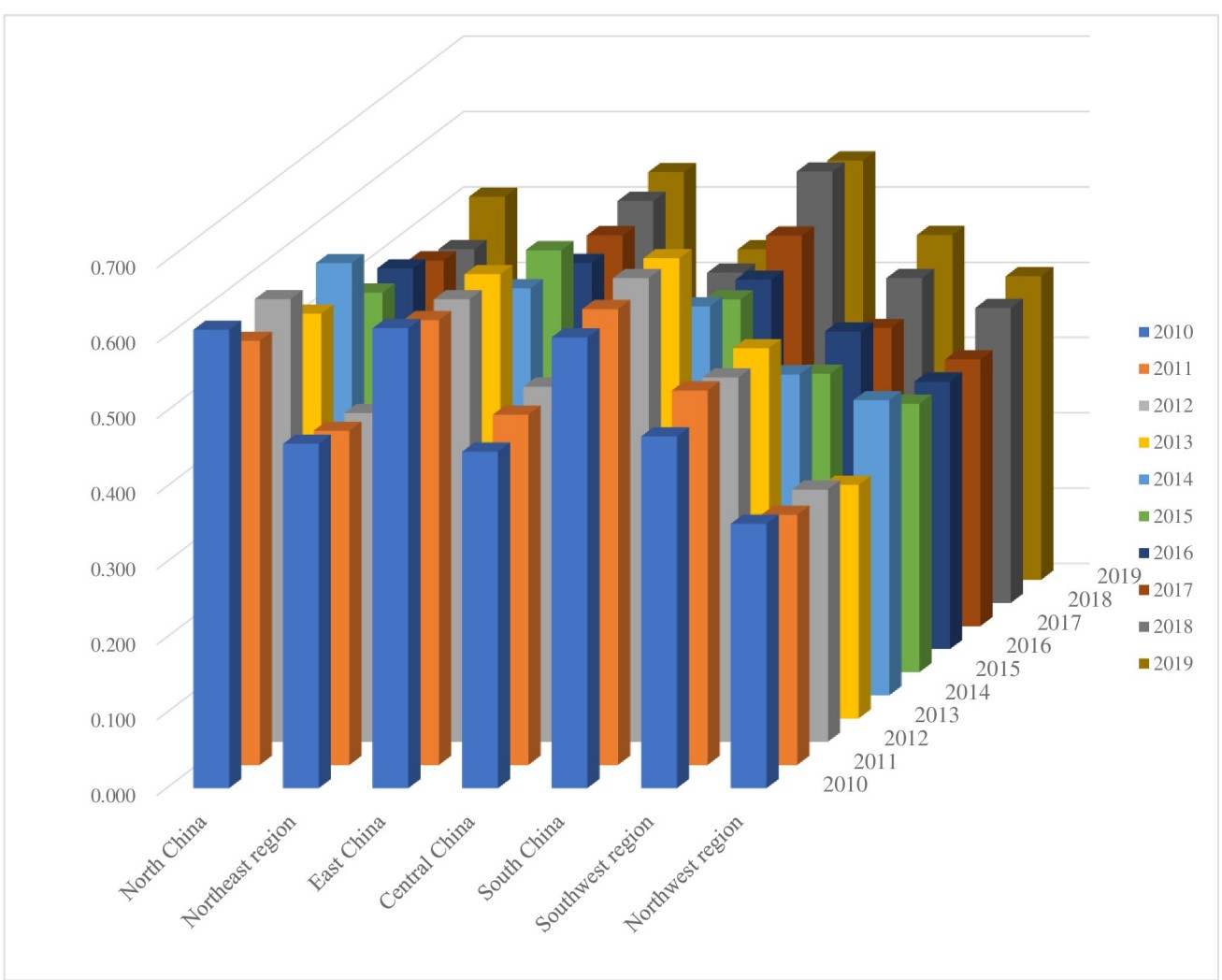

**Fig 3. Changes in the coordination between logistics efficiency and economic development efficiency in different regions of China from 2010 to 2019.**

significant disharmony between logistics efficiency and economic development in Northeast China since 2015. The coordination and coupling degree of logistics efficiency and economic development between Central China and Southwest China was mostly at the basic coordination level during the investigation period. Central China was at a relatively coordinated level in 2015–2017, and Southwest China was at a relatively coordinated level in 2015 and 2017. This indicates that the overall coordination degree of logistics efficiency and economic development in these two regions is high, but there is still room for improvement. The coordination and coupling degree of logistics efficiency and economic development in South China and East China are both in the basic and relatively coordinated level during the inspection period, which indicates that the coordination and coupling degree of logistics efficiency and economic development in these two regions is relatively high. However, in the northwest region, except for 2019, the coordination coupling between logistics efficiency and economic development in other years during the inspection period was at a relatively uncoordinated level, indicating that there is still great room for improvement in the coordination between logistics efficiency and economic development in the northwest region [24].

**Table 6. Regional division based on different coordination levels.**

| Degree of coordination | Serious disharmony | Less coordinated | Basic coordination | Relatively coordinated | Very coordinated |
|---|---|---|---|---|---|
| 2010 | | Northwest China | Northeast China, Central China, South China, Southwest China | North China, East China | |
| 2011 | | Northwest China | North China, Northeast China, East China, Central China, Southwest China | South China, | |
| 2012 | | Northwest China | North China, Northeast China, East China, Central China, Southwest China | South China, | |
| 2013 | | Northwest China | North China, Northeast China, East China, Central China, Southwest China | South China | |
| 2014 | | Northwest China | North China, Northeast China, East China, Central China, Southwest China, South China | | |
| 2015 | | Northwest China, Northeast China, Central China, Southwest China | North China, East China, South China | | |
| 2016 | | Northwest China, Northeast China, Central China | North China, East China, Southwest China, South China | | |
| 2017 | | Northwest China, Northeast China, Central China, Southwest China | North China, East China, South China | | |
| 2018 | | Northwest China, Northeast China | North China, , East China, Central China, Southwest China, South China | | |
| 2019 | | Northeast China | North China, , East China, Central China, Southwest China, South China, Northwest China | | |

To further analyze the relationship between logistics efficiency, economic development, and their coupling degree, a three-dimensional scatter plot is generated as shown in Fig 4, with logistics efficiency score as the x-axis, economic development score as the y-axis, and the value of logistics efficiency and economic development coupling degree as the z-axis. Considering that the values of x, y, and z are all between 0 and 1, 0.5 is used as the origin of the 3D scatter plot.

It can be seen from Fig 4 that there is an apparent positive correlation between logistics efficiency, economic development, and the degree of coupling between the two; that is, in regions with high logistics efficiency and economic development, the interaction between the two is significantly stronger than in the other areas. As shown in Fig 4, South China has always been relatively coupled and coordinated during the survey period (.) The coupling and coordination level between logistics efficiency and economic development in this region is far ahead of other parts of China. In areas with low logistics efficiency and economic development levels, the coupling and coordination effect between the two is weak, and the level of coordinated development could be better. Suppose the northwest region of China has been relatively uncoordinated for most of the time. In that case, it indicates that there is still significant room for improvement in this region's coupling and coordination level of logistics efficiency and economic development.

However, by observing Fig 4, it can also be observed that there is only sometimes a positive correlation between logistics efficiency, economic development, and their coupling degree in China; that is, logistics efficiency and economic development are only sometimes coordinated and consistent. For example, in North China in 2015, the logistics efficiency was high, the economic development was low, and the coordination coupling degree between the two reached the basic coupling level. The Southwest region's logistics efficiency and economic development in 2018 were both relatively low. Still, the coordination and coupling between the two reached a basic coupling level. The deviation between logistics efficiency, economic development level, and coupling coordination degree in some regions has occurred in some years, possibly due to

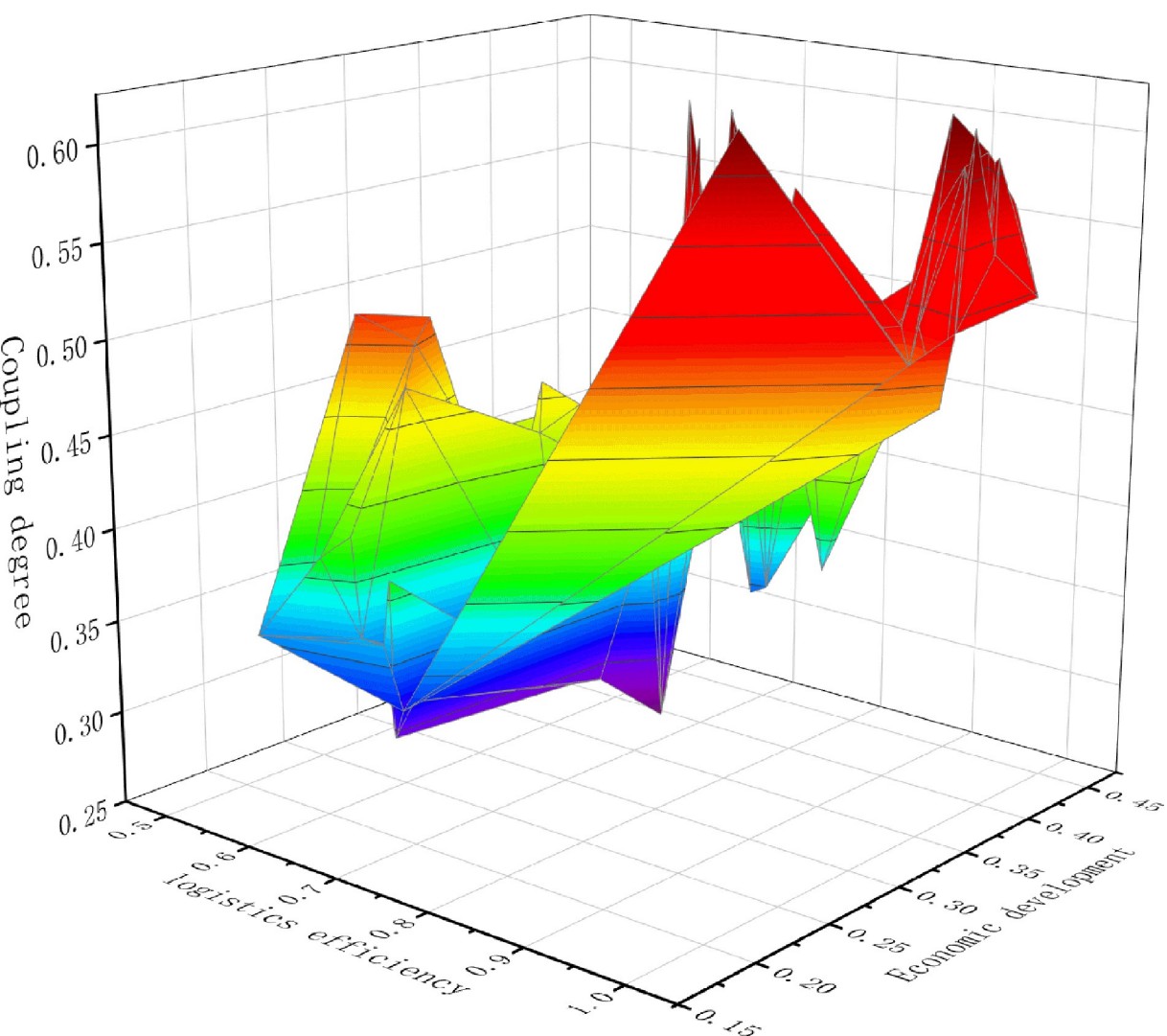

**Fig 4. Scatter plot of the relationship between logistics efficiency, economic development, and their coupling degree.**

structural problems in the development process of the logistics industry in some regions of China. For example, although the scale of logistics is large, the development of the logistics industry relies more on hardware facilities. At the same time, the proposal of dual carbon requirements has brought significant challenges to logistics and economic development in some regions of China.

# VI. Conclusions and development suggestions

## A. Conclusions and contributions

This paper analyzes the spatiotemporal evolution characteristics of the coupling coordination between regional logistics efficiency and economic development in China from 2010 to 2019 by constructing an evaluation index system for regional logistics and economic development, using methods such as cross DEA method and coupling coordination degree model. The research findings can be described as follows,

1. There are significant differences in the overall efficiency development level of logistics in different regions of China, and regional development needs to be more balanced. It shows a pattern of "East China>North China>South China>Central China>Northeast China>Northwest China>Southwest." Specifically, the average logistics efficiency in East China and North China is more than 0.9, and the development level of the logistics industry is high, while the average logistics efficiency in Northwest China and Southwest China is only 0.5–0.6, and the development level of the logistics industry is low.

2. The overall development of China's regional economy shows a fluctuating and stable rise. From a regional perspective, the average levels of high-quality economic development in the seven major regions of North China, Northeast China, East China, Central China, South China, Southwest China, and Northwest China from 2010 to 2019 were 0.389, 0.329, 0.428, 0.377, 0.356, 0.301, and 0.191, respectively. The regional economic development presents significant regional differences, with the development level gradually decreasing from coastal areas such as East China, North China, and South China to inland areas such as Southwest and Northwest.

3. The coordination and coupling between logistics efficiency and economic development in different regions of China exhibit temporal fluctuations and spatial differences. In general, the coordination degree between logistics efficiency and economic development in North China and Northeast China fluctuated from 2010 to 2019, while that in East China fluctuated around 0.6. The coordination coupling degree between logistics efficiency and economic development in Central China and Southwest China is characterized by first rising and then falling. The coordination and coupling degree between logistics efficiency and economic development in South China is relatively high all year round, while the coordination and coupling degree between logistics efficiency and economic development in Northwest China is relatively low all year round.

The main contributions of this paper are threefold: firstly, this paper analyzes the coupling and coordination relationship between logistics efficiency and economic development from a regional perspective, rather than a common provincial perspective, which is also in line with the regionalization characteristics of China's economic development. Secondly, based on existing research, this paper not only considers common indicators that characterize logistics efficiency and economic development, but also incorporates indicators such as energy consumption and carbon emissions when constructing an evaluation index system. This is consistent with the goal of high-quality and green development of China's economy. Thirdly, based on the research results of Ruhe et al. (2022), this paper proposes a calculation method for the coordination coupling degree between regional logistics efficiency and economic development, and provides a division standard for the coupling degree, which can provide a basis for the coupling analysis of regional logistics efficiency and economic development.

## B. Development suggestions

Based on the above analysis and conclusions, this paper puts forward the following policy suggestions for improving the coupling and coordinated development level of regional economic development and logistics development in China.

1. Improve logistics infrastructure construction, strengthen logistics technology innovation and regional cooperation. Increase financial support, continuously improve and innovate investment and financing models in the field of infrastructure, increase the construction of logistics infrastructure, plan the layout of transportation hub nodes and expressways

reasonably, continuously improve the transportation system, and build an efficient and convenient logistics infrastructure network [25]. On the other hand, we should strengthen the application of advanced logistics technology and equipment, promote the reform of the supply side structure of the logistics industry, reduce logistics costs, and improve enterprise information management. At the same time, we will strengthen regional cooperation, promote the co-construction and sharing of transportation hub facilities, strengthen the interconnection between branch lines and main lines, promote the reasonable flow of advanced logistics technology and information exchange between regions, and leverage the "spillover effect" and "division of labor effect" to gradually converge and coordinate the development of regional logistics technology differences.

2. Encouraging enterprises in coastal areas to expand into inland areas in order to promote a balanced distribution of resources and markets. As mentioned earlier, the coastal regions of East China, North China and South China have a high level of regional economic development, reaching 0.428, 0.389 and 0.356, respectively. This is closely related to the logistical advantages of the coastal regions. However, the development level of inland areas in Northwest China is only 0.191. Therefore, there is a need to strengthen cooperation and coordination between coastal and inland areas in order to promote complementary resources and synergistic development, to achieve a balanced and sustainable growth of the national regional economy. In addition, the Government's attention to logistics development should be raised and policies favorable to attracting investment and developing industries in inland areas, such as tax incentives and industrial support policies, should be formulated.

3. Balancing the contradiction between regional logistics efficiency and economic development and promoting the coordinated development of regional logistics efficiency and economic development. According to the previous analysis, logistics efficiency and economic development in South China have maintained a relatively high degree of coordination over time, with an average value of 0.558; however, Northwest China has been relatively low for a long time, with an average value of 0.358. Therefore, it is necessary for Northwest China to deepen further the integration and development of logistics with the industries of agriculture, industry, and services. In addition, other regions can learn from the regional policies and development experience of South China. Still, they must also adapt to local conditions, leverage strengths and avoid weaknesses to seek a development path that suits their respective situations.

## C. Limitations and future direction

This paper mainly uses the cross DEA method to study the coupling relationship between regional logistics efficiency and economic development in China. The empirical analysis results also provide a series of meaningful results, providing a reference basis for better coordinated development of regional logistics efficiency and economic development in China. However, as the results of this study are based on data from the Chinese context, the relevant conclusions may not be fully applicable to other countries. In subsequent research, the coupling relationship between regional logistics efficiency and economic development in different countries can be further compared to enhance the reliability and applicability of the analysis results. In addition, due to limitations in data availability, this study mainly analyzes the coupling relationship between regional logistics efficiency and economic development from a comparative macro level. There has been no in-depth research on how regional logistics efficiency and economic development interact. With the increasing diversity of data acquisition

methods and the continuous development of data mining analysis technology in the future, multi-source data can be used to conduct multi-level and multi-dimensional analysis of the coupling relationship between regional logistics efficiency and economic development, in order to better understand the coupling relationship between regional logistics efficiency and economic development.

## Supporting information

**S1 Data. Data 1–5 are included in the file.**
(ZIP)

## Acknowledgments

N.L. designed the study, conducted data analysis, and drafted the manuscript. N. L. conducted data analyses and contributed to the writing. TX. M and XC. D contributed to the review of the manuscript. All authors contributed to this article and approved the submitted version. The authors thank all the investigators and participants for the National Bureau of Statistics data.

## Author Contributions

**Data curation:** Na Li.

**Formal analysis:** Tianxing Ma.

**Funding acquisition:** Tianxing Ma.

**Investigation:** Na Li.

**Resources:** Xiaochun Deng.

**Software:** Xiaochun Deng.

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
