## [Decision Letter · Decision Letter 0]

24 Jul 2023

PONE-D-23-19917Analysis of the Coupling Degree between Regional Logistics Efficiency and Economic Development CoordinationPLOS ONE

Dear Dr. li,

Thank you for submitting your manuscript to PLOS ONE. After careful consideration, we feel that it has merit but does not fully meet PLOS ONE’s publication criteria as it currently stands. Therefore, we invite you to submit a revised version of the manuscript that addresses the points raised during the review process.

We look forward to receiving your revised manuscript.

Kind regards,

Mohamed Rafik N Qureshi, Ph.D.

Academic Editor

PLOS ONE

“This study was supported by Soft Science Research of Sichuan Provincial Department of Science and Technology (No. 2019JDR0043) and Guangxi Natural Science Foundation Project (No. 2018JJB180015).”

“no”

Additional Editor Comments:

The manuscript "Analysis of the Coupling Degree between Regional Logistics Efficiency and Economic Development Coordination" should undergo "Major Revision" and needs to be modified as per the reviewers' comments.

Reviewers' comments:

Reviewer's Responses to Questions

**Comments to the Author**

1. Is the manuscript technically sound, and do the data support the conclusions?

Reviewer #1: Partly

Reviewer #2: Partly

2. Has the statistical analysis been performed appropriately and rigorously? 

Reviewer #1: N/A

Reviewer #2: Yes

3. Have the authors made all data underlying the findings in their manuscript fully available?

Reviewer #1: Yes

Reviewer #2: Yes

4. Is the manuscript presented in an intelligible fashion and written in standard English?

Reviewer #1: No

Reviewer #2: Yes

5. Review Comments to the Author

Reviewer #1: I hope these comments are useful, and I wish you all the best in further developing your research.

1. The data covers 2010-2019, now is 2023, the data seems a bit old, and which can be updated to at least 2021.

2. There are many grammatical mistakes in the manuscript and the English should be improved.

3. The number of references is not enough.

Reviewer #2: This study discuss the Logistics Efficiency and Economic Development Coordination. During review, I found that this article required revision. Following are the comments:

1. Abstract should be refined and emphasized on the key findings of the study.

2. Introduction section is not properly highlights the objective of the study and research gap. I recommend authors to clearly include the "Research objective".

3. Literature review sections are poorly developed. Therefore, I suggest authors to rework on this section and include the recently published articles. Following articles can be considered but not necessary:

https://doi.org/10.1002/sd.2266

https://doi.org/10.1007/s11356-021-16961-1

4. Methodology is not clearly explain all the steps of this research approach.

5. Conclusion is inconclusive. Please expand this section. Also, i suggest authors to properly write the theoretical and practical implications aligned with the research objective.

6. English language needs to be improve through proofreading.

6. PLOS authors have the option to publish the peer review history of their article (what does this mean?). If published, this will include your full peer review and any attached files.

Reviewer #1: No

Reviewer #2: No

---

## [Author Response · Author response to Decision Letter 0]

31 Aug 2023

Dear reviewer,

We thank the reviewers for providing constructive feedback. We have fully revised our manuscript (PONE-D-23-19917)and have addressed all of the reviewers' comments, as well as added new analyses to further strengthen our work. 

Based on your comments and requests, we have made extensive revisions to the original manuscript. We are attaching the revised manuscript in both PDF and MS word format for your review. Also, we have attached a Point-by-point response to reviewers' file. We have also attached a revised draft with the changes highlighted in a Yellow highlighted background as additional material for checking/editing.

Reviewer #1: I hope these comments are useful, and I wish you all the best in further developing your research.

1.1 The data covers 2010-2019, now is 2023, the data seems a bit old, and which can be updated to at least 2021. 

Response 1.1#

Thank you for the questions raised by the reviewers. The reason why the data in this paper were selected from 2010 to 2019 is that the COVID-19 pandemic at the end of 2019 had a significant impact on economic activities. In particular, China's adherence to the dynamic Zero-COVID has had a particularly serious impact on economic activities. As a result, there is a significant deviation in the economic data of recent years compared to the data before 2019. Therefore, in order to avoid the possible distortion impact of data on the overall analysis after 2019, this article only selects data from 2010 to 2019 for analysis. However, this does not affect the conclusion of this article regarding the coordination and coupling between regional logistics efficiency and economic development. In addition, in subsequent research, the author of this article will also adopt the opinions of reviewers and use the latest data for research.

1.2 There are many grammatical mistakes in the manuscript and the English should be improved.

Response 1.2#

Every effort has been made to improve the manuscript and some changes have been made. These changes do not affect the content or framework of the paper. Instead of listing these changes, we have highlighted them in blue in the revised paper. We would like to express our sincere gratitude to reviewer #1 for his enthusiastic work and hope that the revised grammatical structure will be recognized.

1.3 The number of references is not enough.

Response 1.3#

As suggested by the commenter, we have added additional references to support this main point (Refs.1.2.3……9). The references are listed below and have now been added to the corresponding reference positions in the manuscript [17-25].

1. Khan, S.A.R., Yu, Z. & Umar, M. A road map for environmental sustainability and green economic development: an empirical study. Environ Sci Pollut Res 29, 16082–16090 (2022).

2. Khan S A R, Yu Z, Umar M, et al. Renewable energy and advance- d logistical infrastructure: Carbon‐free economic development[J]. Sustainable Development, 2022, 30.

3. Jun-Sheng L, Yao-Feng M A, Bing W U. The dynamic analysis of spatial and temporal differences of the coupling coordination degree about the inbound tourists flows and regional economy: * based on the regional panel data of 31 provinces from 1993 to 2011[J]. Economic Management Journal, 2015.

4. Liang Z, Chiu Y H, Guo Q, et al. Low-carbon logistics efficiency: Analysis on the statistical data of the logistics industry of 13 cities in Jiangsu Province, China[J]. Research in transportation business and management, 2022.

5. Xing L, Xue M, Hu M. Dynamic simulation and assessment of the coupling coordination degree of the economy–resource–environment system: Case of Wuhan City in China[J].Journal of Environmental Management, 2019, 230(JAN.15):474-487.DOI:10.1016/j.jenvman.2018.09.065

6. Barilla D, Carlucci F, Cirà, Andrea,et al.Total factor logistics productivity: A spatial approach to the Italian regions[J].Transportation Research Part A: Policy and Practice, 2020, 136.DOI:10.1016/j.tra.2020.03.033.

7. Gvozdenac-Urosevic B. ENERGY EFFICIENCY AND GDP[J].Thermal Science, 2010, 14(3):6-6.DOI:10.2298/TSCI100505006G.

8. Zhang X, Sun Y, Sun Y. Research on Cold Chain Logistics Traceability System of Fresh Agricultural Products Based on Blockchain[J].Computational intelligence and neuroscience, 2022, 2022:1957957.DOI:10.1155/2022/1957957.

9. Niu B, Dai Z, Liu Y, et al.The role of Physical Internet in building trackable and sustainable logistics service supply chains: A game analysis[J].International Journal of Production Economics, 2022, 247.

Reviewer #2: This study discuss the Logistics Efficiency and Economic Development Coordination. During review, I found that this article required revision. Following are the comments:

2.1. Abstract should be refined and emphasized on the key findings of the study.

Response 2.1#

Thank you for the suggestions. We have revised the abstract and emphasized the main research findings. The key findings of this paper are as follows:

The research findings indicate that Eastern and Northern regions of China show higher logistics efficiency, while Northwestern and Southwestern regions exhibit lower logistics efficiency. Coastal areas have relatively higher economic development levels compared to inland areas. Regarding the coupling coordination between logistics efficiency and economic development, different regions show temporal fluctuations and spatial disparities. Some regions demonstrate higher coordination between logistics efficiency and economic development, while others show lower coordination. Additionally, as the economy experiences rapid growth, logistics efficiency also improves, but the level of coordination varies among different provinces.

2.2. Introduction section is not properly highlights the objective of the study and research gap. I recommend authors to clearly include the "Research objective".

Response 2.2#

Thank you for the suggestions. We have revised the introduction section based on your suggestions.

The research objective of this paper is:

However, in practical situations, some regions may face issues such as incomplete logistics networks, outdated transportation methods, and inadequate logistics information flow, leading to low logistics efficiency and impacting the speed and quality of economic development. On the other hand, the varying levels of economic development also impose higher demands on the logistics system, requiring it to adapt and support faster and more flexible economic growth. As a result, there are significant differences in logistics development and economic development among different regions. These disparities may influence the coordination and coupling degree between logistics efficiency and economic development, necessitating a regional-specific analysis to develop more targeted optimization measures.

The research gap of this paper is:

Currently, research mainly focuses on the relationship between logistics efficiency of the logistics industry or companies and economic development, but there is less exploration from a regional perspective regarding the connection between logistics development and economic growth. Additionally, when selecting evaluation indicators, researchers usually consider only internal influencing factors such as freight volume and goods turnover. However, the coordination between regional logistics efficiency and economic development is not only influenced by internal factors but also affected by external and regulatory factors. The existing research lacks consideration of the impact of external indicators such as energy consumption and carbon emissions on logistics development and economic growth.

2.3. Literature review sections are poorly developed. Therefore, I suggest authors to rework on this section and include the recently published articles. Following articles can be considered but not necessary:

Response 2.3#

We sincerely appreciate your valuable comments. We have carefully reviewed the literature and have added https://doi.org/10.1002/sd.2266 and

https://doi.org/10.1007/s11356-021-16961-1 literature, and refer to the above literature and improve this section.

2.4. Methodology is not clearly explain all the steps of this research approach.

Response 2.4#

Thank you for pointing out the issue. In the first paragraph of Section 3, we have provided a supplementary explanation of the methodology used in this paper.

This paper adopts the DEA cross-efficiency model and the coupling coordination degree calculation formula to comprehensively evaluate the efficiency and coordination degree of regional logistics and economic development. The DEA cross-efficiency model quantitatively assesses the efficiency of regional logistics and economic development using self-assessment values and cross-efficiency as comprehensive indicators. The coupling coordination degree calculation formula considers the cross-efficiency values of logistics and economic development to measure the coordination degree between logistics and economic development. The specific steps of the evaluation method are as follows: first, applying the DEA cross-efficiency model to calculate the efficiency level of regional logistics and economic development, and then using the coupling coordination degree calculation formula to calculate the coordination degree between logistics and economic development.

2.5. Conclusion is inconclusive. Please expand this section. Also, i suggest authors to properly write the theoretical and practical implications aligned with the research objective.

Response 2.5#

Thank you to the reviewers for their feedback. In response to expert opinions, the author has supplemented, expanded, and improved the conclusion section. The main modifications (highlighted in yellow) are as follows：

VI. Conclusions and development suggestions 

A. Conclusions and contributions

This paper analyzes the spatiotemporal evolution characteristics of the coupling coordination between regional logistics efficiency and economic development in China from 2010 to 2019 by constructing an evaluation index system for regional logistics and economic development, using methods such as cross DEA method and coupling coordination degree model. The research findings can be described as follows.

The main contributions of this paper are threefold: firstly, this paper analyzes the coupling and coordination relationship between logistics efficiency and economic development from a regional perspective, rather than a common provincial perspective, which is also in line with the regionalization characteristics of China's economic development. Secondly, based on existing research, this paper not only considers common indicators that characterize logistics efficiency and economic development, but also incorporates indicators such as energy consumption and carbon emissions when constructing an evaluation index system. This is consistent with the goal of high-quality and green development of China's economy. Thirdly, based on the research results of Ruhe et al. (2022), this paper proposes a calculation method for the coordination coupling degree between regional logistics efficiency and economic development, and provides a division standard for the coupling degree, which can provide a basis for the coupling analysis of regional logistics efficiency and economic development.

B. Development suggestions

Based on the above analysis and conclusions, this paper puts forward the following policy suggestions for improving the coupling and coordinated development level of regional economic development and logistics development in China. 

C. Limitations and future direction

This paper mainly uses the cross DEA method to study the coupling relationship between regional logistics efficiency and economic development in China. The empirical analysis results also provide a series of meaningful results, providing a reference basis for better coordinated development of regional logistics efficiency and economic development in China. However, as the results of this study are based on data from the Chinese context, the relevant conclusions may not be fully applicable to other countries. In subsequent research, the coupling relationship between regional logistics efficiency and economic development in different countries can be further compared to enhance the reliability and applicability of the analysis results. In addition, due to limitations in data availability, this study mainly analyzes the coupling relationship between regional logistics efficiency and economic development from a comparative macro level. There has been no in-depth research on how regional logistics efficiency and economic development interact. With the increasing diversity of data acquisition methods and the continuous development of data mining analysis technology in the future, multi-source data can be used to conduct multi-level and multi-dimensional analysis of the coupling relationship between regional logistics efficiency and economic development, in order to better understand the coupling relationship between regional logistics efficiency and economic development.

2.6. English language needs to be improve through proofreading.

Response 2.6#

We have tried our best to improve the manuscript and have made some finishing touches to the manuscript. These changes do not affect the content and framework of the paper. Instead of listing these changes here, we have shown them in blue font in the revised paper. Once again, we would like to express our sincere thanks to reviewer #2 for his enthusiastic work and hope that the revisions will be recognised.

Thank you very much for your attention and valuable time. Looking forward to your replies. I also wish you all the best in your endeavors and wish you all the best.

Yours sincerely, Na-Li

9 August 2023

---

## [Decision Letter · Decision Letter 1]

14 Sep 2023

PONE-D-23-19917R1Analysis of the Coupling Degree between Regional Logistics Efficiency and Economic Development CoordinationPLOS ONE

Dear Dr. li,

Thank you for submitting your manuscript to PLOS ONE. After careful consideration, we feel that it has merit but does not fully meet PLOS ONE’s publication criteria as it currently stands. Therefore, we invite you to submit a revised version of the manuscript that addresses the points raised during the review process.

We look forward to receiving your revised manuscript.

Kind regards,

Mohamed Rafik N. Qureshi, Ph.D.

Academic Editor

PLOS ONE

Journal Requirements:

Additional Editor Comments:

Thank you for revising the manuscript entitled "Analysis of the Coupling Degree between Regional Logistics Efficiency and Economic Development Coordination". The reviewers' new comments may be addressed.

Reviewers' comments:

Reviewer's Responses to Questions

**Comments to the Author**

1. If the authors have adequately addressed your comments raised in a previous round of review and you feel that this manuscript is now acceptable for publication, you may indicate that here to bypass the “Comments to the Author” section, enter your conflict of interest statement in the “Confidential to Editor” section, and submit your "Accept" recommendation.

Reviewer #1: All comments have been addressed

Reviewer #2: (No Response)

2. Is the manuscript technically sound, and do the data support the conclusions?

Reviewer #1: Yes

Reviewer #2: (No Response)

3. Has the statistical analysis been performed appropriately and rigorously? 

Reviewer #1: Yes

Reviewer #2: (No Response)

4. Have the authors made all data underlying the findings in their manuscript fully available?

Reviewer #1: Yes

Reviewer #2: (No Response)

5. Is the manuscript presented in an intelligible fashion and written in standard English?

Reviewer #1: Yes

Reviewer #2: (No Response)

6. Review Comments to the Author

Reviewer #1: Overall, the current paper is complete and clear, but there are still some paragraphs that can be improved.

1. In III. Evaluation method, you may further clarify why the DEA cross efficiency model is suitable for solving the research questions in this paper.

2. Region is a relative concept that can be large or small areas. The background of such regions is better to be briefly introduced, so that international readers can also know about it.

3. Development suggestions (2)and (3) went too far beyond the results of the paper, and they are empty, needing to focus on the topic and results of the paper so as to provide some more practical suggestions.

Reviewer #2: This article is ready for publishing.

7. PLOS authors have the option to publish the peer review history of their article (what does this mean?). If published, this will include your full peer review and any attached files.

Reviewer #1: No

Reviewer #2: No

---

## [Author Response · Author response to Decision Letter 1]

23 Sep 2023

Dear Editors and Reviewers,

 Thank you for your letter and for the reviewers’ comments concerning our manuscript entitled “Analysis of the Coupling Degree between Regional Logistics Efficiency and Economic Development Coordination” (ID: PONE-D-23-19917). Those comments are all valuable and very helpful for revising and improving our paper, as well as the important guiding significance to our researches. We have studied comments carefully and have made correction which we hope meet with approval.

 Based on your comments and requests, we have made extensive revisions to the original manuscript. We are attaching the revised manuscript in both PDF and MS word format for your review. Also, we have attached a Point-by-point response to reviewers' file. We have also attached a revised draft with the changes highlighted in a Yellow highlighted background as additional material for checking/editing.

 

Reviewer # 1: Overall, the current paper is complete and clear, but there are still some paragraphs that can be improved.

Question 1.1. In III. Evaluation method, you may further clarify why the DEA cross efficiency model is suitable for solving the research questions in this paper.

Response 1.1 #

Thank you for your suggestion. Firstly, indeed there was an omission in this section and we apologize for not clarifying the reasons for using the DEA cross-efficiency model in the original article; 

Secondly, in order to reflect the reviewer's concerns more clearly, we have explained the DEA cross-efficiency model in more detail, as shown below: 

 Doyel and Green (1994) proposed a DEA cross-efficiency model to solve DEA decision units' sorting and comparison problems effectively. This model goes beyond the scope of only considering individual selection preferences of DMUs but comprehensively incorporates factors that affect the weight changes of other DMUs. By using self-evaluation values and cross-efficiency as comprehensive indicators to measure the efficiency of DMUs, the cross-efficiency model solves the problem of incompatibility caused by inconsistent input and output weights in the CCR model. Based on that, this paper uses the DEA cross-efficiency to take into account the interactions between economic development and logistics efficiency among these regional areas.

Lastly, we have supplemented it in the original article III. Evaluation Methods and highlighted it in yellow.

Question 1.2. Region is a relative concept that can be large or small areas. The background of such regions is better to be briefly introduced, so that international readers can also know about it.

Response 1.2 #

We are grateful for the suggestion. To be more clearly and in accordance with the reviewer concerns, we have added a more detailed interpretation regarding region. As shown in the following: 

A region usually refers to a certain geographical space. It has a certain area, shape, scope or boundary. Due to the vastness of China's territory, there are various types of geographic regions analyzed from multiple perspectives: topographical, climatic, human, economic and political. In general, China is usually divided into seven geographic regions. This paper focuses on the degree of coordinated coupling between regional logistics efficiency and economic development, with economic geography as the primary basis, so this paper follows this regional division into the following seven regions：

At the same time, we have supplemented it in the original article IV. Evaluation indicators and data sources （B. Data sources） and highlighted it in yellow.

Question 1.3. Development suggestions (2) and (3) went too far beyond the results of the paper, and they are empty, needing to focus on the topic and results of the paper so as to provide some more practical suggestions.

Response 1.3 #

Thank you for your suggestion. We agree with your comments and in order to better present some more practical suggestions, we have rewritten this text in the revised version as follows and highlighted it in yellow:

(2) Encouraging enterprises in coastal areas to expand into inland areas in order to promote a balanced distribution of resources and markets. As mentioned earlier, the coastal regions of East China, North China and South China have a high level of regional economic development, reaching 0.428, 0.389 and 0.356, respectively. This is closely related to the logistical advantages of the coastal regions. However, the development level of inland areas in Northwest China is only 0.191. Therefore, there is a need to strengthen cooperation and coordination between coastal and inland areas in order to promote complementary resources and synergistic development, to achieve a balanced and sustainable growth of the national regional economy. In addition, the Government's attention to logistics development should be raised and policies favorable to attracting investment and developing industries in inland areas, such as tax incentives and industrial support policies, should be formulated.

(3) Balancing the contradiction between regional logistics efficiency and economic development and promoting the coordinated development of regional logistics efficiency and economic development. According to the previous analysis, logistics efficiency and economic development in South China have maintained a relatively high degree of coordination over time, with an average value of 0.558; however, Northwest China has been relatively low for a long time, with an average value of 0.358. Therefore, it is necessary for Northwest China to deepen further the integration and development of logistics with the industries of agriculture, industry, and services. In addition, other regions can learn from the regional policies and development experience of South China. Still, they must also adapt to local conditions, leverage strengths and avoid weaknesses to seek a development path that suits their respective situations.

Reviewer # 2: This article is ready for publishing.

Response 2 #

 On behalf of all the contributing authors, I would like to express my sincere gratitude for your acceptance and recognition of our articles.

Many thanks to all the reviewers and editors for their attention and valuable time. I look forward to your replies. I also wish you all the best and all the best.

Yours sincerely, Na Li

Sept. 21, 2023

---

## [Decision Letter · Decision Letter 2]

9 Oct 2023

Analysis of the Coupling Degree between Regional Logistics Efficiency and Economic Development Coordination

PONE-D-23-19917R2

Dear Dr. li,

We’re pleased to inform you that your manuscript has been judged scientifically suitable for publication and will be formally accepted for publication once it meets all outstanding technical requirements.

Kind regards,

Mohamed Rafik N. Qureshi, Ph.D.

Academic Editor

PLOS ONE

Additional Editor Comments (optional):

The manuscript entitled "Analysis of the Coupling Degree between Regional Logistics Efficiency and Economic Development Coordination" has been modified as per the reviewers' comments.

Reviewers' comments:

Reviewer's Responses to Questions

**Comments to the Author**

1. If the authors have adequately addressed your comments raised in a previous round of review and you feel that this manuscript is now acceptable for publication, you may indicate that here to bypass the “Comments to the Author” section, enter your conflict of interest statement in the “Confidential to Editor” section, and submit your "Accept" recommendation.

Reviewer #1: All comments have been addressed

Reviewer #2: (No Response)

2. Is the manuscript technically sound, and do the data support the conclusions?

Reviewer #1: Yes

Reviewer #2: (No Response)

3. Has the statistical analysis been performed appropriately and rigorously? 

Reviewer #1: Yes

Reviewer #2: (No Response)

4. Have the authors made all data underlying the findings in their manuscript fully available?

Reviewer #1: Yes

Reviewer #2: (No Response)

5. Is the manuscript presented in an intelligible fashion and written in standard English?

Reviewer #1: Yes

Reviewer #2: (No Response)

6. Review Comments to the Author

Reviewer #1: (No Response)

Reviewer #2: Authors incorporated all the raised comments.

7. PLOS authors have the option to publish the peer review history of their article (what does this mean?). If published, this will include your full peer review and any attached files.

Reviewer #1: No

Reviewer #2: No

---

## [Editor Report · Acceptance letter]

12 Oct 2023

PONE-D-23-19917R2 

Analysis of the Coupling Degree between Regional Logistics Efficiency and Economic Development Coordination 

Dear Dr. Li:

I'm pleased to inform you that your manuscript has been deemed suitable for publication in PLOS ONE. Congratulations! Your manuscript is now with our production department. 

Kind regards, 

on behalf of

Prof.(Dr.) Mohamed Rafik N. Qureshi 

Academic Editor

PLOS ONE